# Lived experience of people with lateral elbow tendinopathy: a qualitative study from the OPTimisE pilot and feasibility trial

Marcus Bateman [ORCID],[1] Jonathan C Hill [ORCID],[2] Karin Cooper,[3] Chris Littlewood,[4] Benjamin Saunders [ORCID] [2]

[1]Derby Shoulder Unit, University Hospitals of Derby and Burton NHS Foundation Trust, Derby, UK
[2]School of Medicine, Keele University, Keele, UK
[3]Patient Representative, Derby, UK
[4]Faculty of Health, Social Care & Medicine, Edge Hill University, Ormskirk, UK

**Correspondence to**
Mr Marcus Bateman;
marcus.bateman@nhs.net

## ABSTRACT

**Objectives** To explore the lived experience of people with lateral elbow tendinopathy (LET) and its impact on everyday life.

**Design** Qualitative semi-structured interviews, analysed using thematic analysis.

**Setting** Conducted as part of the mixed-methods OPTimisE pilot and feasibility randomised controlled trial of outpatient physiotherapy patients in the UK.

**Participants** 17 participants with LET, purposively sampled from the trial to provide representativeness based on age, sex, ethnicity, deprivation index and treatment allocation.

**Results** Four themes were identified from the participants' responses: (1) cause of onset—typically symptoms were attributed to: sudden changes in activity, repetitive work or compensating for other musculoskeletal conditions; (2) impact on everyday life—which included substantial impacts on quality-of-life, particularly due to pain disturbing sleep and difficulties performing daily tasks (related to work and hobbies) due to pain, although most reported being able to persevere with work; (3) self-help and understanding of the condition—with uncertainty about the appropriateness and potential harm of online advice and confusion from the diagnostic term 'Tennis Elbow' that non-sporting individuals struggled to relate to; (4) healthcare experiences—the treatments received were highly variable and often perceived as ineffectual.

**Conclusions** For the first time, the lived experience of people from a range of backgrounds suffering from LET has been explored. Findings suggest that people frequently related the cause to a specific activity. They reported substantial impacts on daily tasks, sleep, work and hobbies. People also reported hesitancy to trust online information without formal healthcare advice, were confused by the common label of 'Tennis Elbow', and perceived the wide array of healthcare treatment options they had received to offer false hope and be largely ineffective. This study provides stimulus for clinicians to consider the advice and treatment provided, and whether the messages conveyed reflect the favourable natural history of the condition.

**Trial registration number** ISRCTN64444585

## STRENGTHS AND LIMITATIONS OF THIS STUDY

⇒ This qualitative study explores the impact of lateral elbow tendinopathy (LET) on everyday life from the perspective of the individual.

⇒ Individuals from a range of socioeconomic and ethnic backgrounds contributed their views and opinions.

⇒ We acknowledge that qualitative studies draw on the opinions of those interviewed and may not be representative of the wider population; however, our aim was in-depth exploration of these views.

⇒ The opinions presented are from people accessing healthcare for LET and so may not reflect the views of those who do not access healthcare for LET, due to lower symptom severity, stoicism or inaccessibility.

## INTRODUCTION

Lateral elbow tendinopathy (LET) is a painful elbow condition, commonly known as Tennis Elbow. The term Tennis Elbow is thought to date back to 1882, when the condition was originally described as Lawn Tennis Arm, and has been used ever since.[1] Only recently, in 2019, has there been a call for this description to be changed to LET following an international consensus exercise involving clinicians and researchers.[2] However, the views of patients were not considered when proposing this change in terminology.

LET describes a painful condition involving the extensor tendons of the forearm near the lateral epicondyle of the elbow, or enthesis, from where they originate. It occurs frequently, with point prevalence in the adult general population at a given time, reported between 1.3% and 4.4%.[3 4] While the impact of the condition has been studied in relation to work absence and healthcare usage, there are no studies to our knowledge that explore the experience of living with the condition, termed 'the lived experience' for the purpose

of this study.[5–7] By understanding the perceptions of people living with the condition, clinicians may be better-equipped to offer advice and education to help them cope, as for many the symptoms will resolve naturally over time.[8]

This qualitative study was embedded within a two-arm multi-centred pilot and feasibility randomised controlled trial investigating whether a newly designed optimised physiotherapy protocol (OPTimisE intervention) could be tested against usual care in a real world healthcare setting.[9] In addition to quantitative measures of feasibility, trial participants were interviewed to explore their views and experiences related to the trial design and intervention protocol. The overall aim of the qualitative study was to assess acceptability of the OPTimisE intervention and identify ways in which the method could be improved in preparation for a main trial. Within these interviews, patients described their individual journeys. Although not directly related to the original pilot and feasibility trial aims, the emergent analytic direction provided accounts of the lived experience of people with LET that have not previously been reported. We therefore, aimed in this study, to understand the lived experiences of people with LET that emerged from these in-depth interviews.

## METHOD
### Participant selection
Patients consenting for the OPTimisE pilot and feasibility trial were asked whether they gave permission to be contacted for an individual interview, following their course of physiotherapy treatment. Those who gave permission were purposively sampled to include people with varied ages, gender, ethnicity, deprivation index and treatment allocation within the trial, as far as was possible within the sample recruited to the trial. Patients were sent a letter of invitation by post, accompanied by a participant information sheet and followed up by email or telephone 2 weeks later, to ask if they wished to be interviewed.

### Data collection
Participants were given the option of face-to-face, telephone or video conference calls at a mutually convenient time, but all opted for telephone interviews. All interviews were audio-recorded and participants provided recorded verbal consent after being read a consent form. Participants were sent a £20 gift voucher to thank them for their time.

All interviews were conducted by MB, a male consultant physiotherapist who has qualitative research training. The interviewer was not known to the participants but they were aware that he was the chief investigator for the OPTimisE pilot and feasibility trial. Participants were encouraged to speak freely about their opinions, whether positive or negative.

Interviews were semi-structured, using a topic guide developed by MB and BS (see online supplemental file 1) and reviewed by the patient coinvestigator (KC). The topic guide was iteratively revised based on early analysis. Sixty minutes were allocated for each interview but the mean duration was 28 min (range 18–42). Interviews were not repeated.

### Data analysis
Following the interviews, the recordings were uploaded via an encrypted web portal to an independent transcription service (https://www.universitytranscriptions.co.uk/) to be transcribed verbatim and returned via encrypted download. All transcriptions were checked for accuracy by MB and any uncertainties were resolved by relistening to the original audio recording. Transcripts were not returned to participants as there were no data quality issues. Anonymised interview transcripts were analysed using inductive thematic analysis.[10] MB coded all of the transcripts using NVivo V.12 software. Codes were explored both within and across interview transcripts, then indexed into areas of relevance, based on patterns within the data, to form a provisional codebook. MB, BS, JCH and KC then met in person to review the data, finalise the codebook and agree the themes. The pilot trial qualitative objectives originally focused on exploring the acceptability of the trial recruitment methods and intervention delivery. However, as important additional themes emerged relating to patients' lived experience of LET in the first few interviews, these were explored further during subsequent interviews. This paper reports the themes related to patients' lived experiences and the trial acceptability themes are reported elsewhere. Interviews continued until data saturation was reached, assessed in terms of 'informational redundancy', the point at which additional data no longer offers new insights.[11] In what follows we outline the characteristics of the participant sample, before reporting the key themes.

### Patient and public involvement
Patients were involved with the development of the research question, application for grant funding, design of the OPTimisE intervention and design of the mixed-methods pilot and feasibility trial. KC is a member of the OPTimisE Patient and Public Involvement Group, contributing to the trial design, trial management, analysis of the qualitative data and writing of this report. The findings from this study will be disseminated publicly via social media and trial website.

## RESULTS
From a total of 50 participants recruited to the OPTimisE pilot and feasibility trial, 45 gave permission to be contacted to discuss taking part in a qualitative interview. Twenty-four of these patients were invited to be interviewed and 17 participated. One other patient initially agreed to be interviewed but later changed their mind due to busy work and personal schedules. The other six did not respond to email and telephone follow-up. The median age of participants was 47 (range 37–62) with

**Table 1** Participant demographics

| Identifier | Age | Gender |
|---|---|---|
| BHX003 | 47 | Male |
| BHX004 | 47 | Male |
| DER001 | 52 | Male |
| DER002 | 39 | Female |
| DER003 | 54 | Male |
| DER004 | 55 | Female |
| DER006 | 39 | Female |
| DER008 | 54 | Male |
| DER011 | 40 | Female |
| SHE001 | 42 | Female |
| SHE004 | 52 | Female |
| SHE005 | 48 | Female |
| SHE011 | 37 | Male |
| SHE013 | 62 | Male |
| SHE014 | 43 | Female |
| SHE016 | 54 | Male |
| SHE018 | 47 | Male |

an even split related to sex and treatment group allocation within the trial. Individuals from a range of ethnic and social backgrounds were included, representative of the demographic of the general population. Thirteen

identified as white British, one white other, one Pakistani, one Sri Lankan and one Kosovar. Median deprivation score was 6 (range 1–10), where 1 is the highest level of deprivation and 10 is the lowest level of deprivation, measured in deciles. Median symptoms duration was 6 months (range 2–36) and median baseline Patient Reported Tennis Elbow Evaluation (PRTEE) score was 47 (range 18.5–93). The PRTEE is a measure of pain and function at the time of recruitment to the OPTimisE trial. The scale ranges from 0 to 100, with 100 being the highest level of pain and functional impairment. Demographic data is provided in table 1.

Four themes were identified from the data: (1) the cause of onset, (2) impact on everyday life, (3) self-help and understanding of the condition and (4) the healthcare provision that participants had received. The coding tree is displayed in figure 1.

## Theme 1: the cause of onset

Participants related the cause of their symptoms to one of three overlapping scenarios: a sudden change of activity, repetitive activity or from changes in normal physical exertion on the elbow because they were compensating for other health conditions. Sudden changes in activity were sometimes related to social hobbies:

There's one machine at the gym I believe triggered it yes … It was a sitting down apparatus where you take

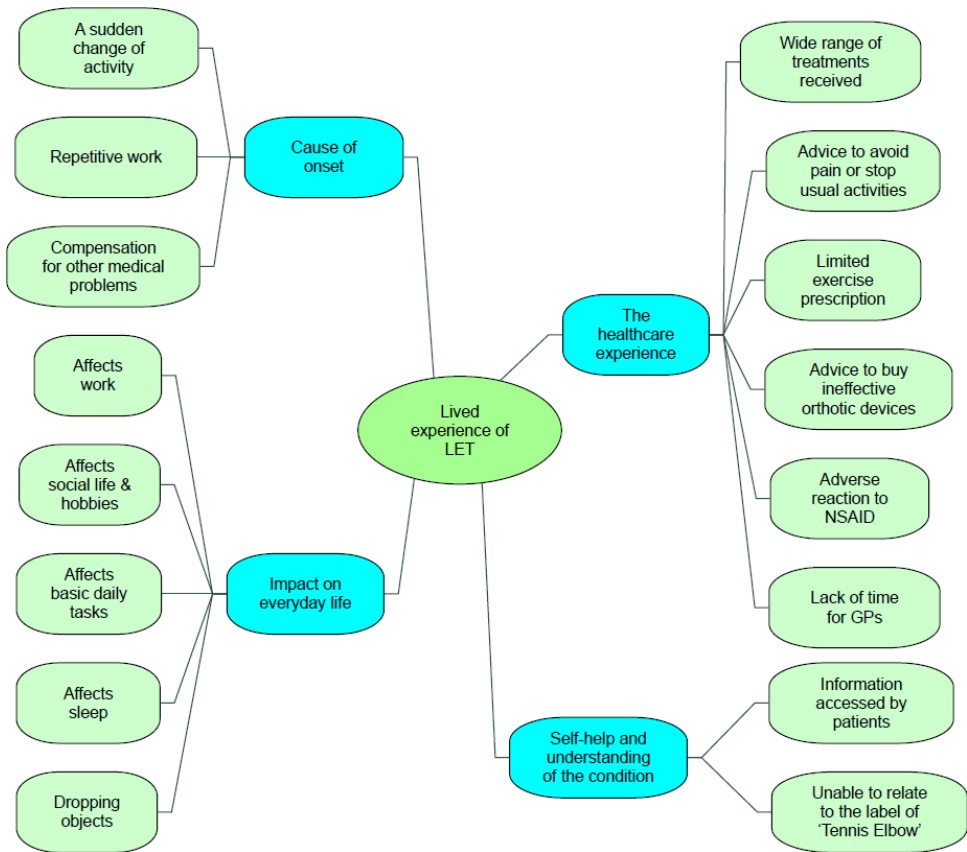

**Figure 1** Coding tree. GP, general practitioner; LTE, lateral elbow tendinopathy; NSAID, non-steroidal anti-inflammatory drug.

your arm, your both arms up into the air and pull the apparatus down … But it's one that I didn't use before, it's not your normal one, it's far apart so, I think that's what triggered it … Yes, I've never done that one, that particular one before, it was a different, just a completely different movement going down. And actually, it didn't feel right from the very first time I did it but I just carried on stupidly. BHX003

The thing which definitely caused Tennis Elbow was a very big gardening project for my mother. She was basically having a concrete wall recast. I've got like some basic construction skills and was doing that with two other people. And it was a big job. We moved something like eight tonnes of material to make it. But because I'd gone from an office job to doing loads of shovelling over an intense period, I think I might have bumped my elbow as well while doing the job. Like I finished at the end of it and like my arm was screaming, right arm. SHE011

Alternatively, sometimes changes in activity were as a result of changes in employment:

SHE014: I think probably because my job is different to how it was previously when I first had it. So, whereas now I'm pretty much full time at a desk, so I'm constantly typing.

MB: Right, okay.

SHE014: So, I think that probably didn't help and I probably wasn't able to rest it as much as I may have done when I had it previously.

MB: Okay. Were you not behind the desk all the time before?

SHE014: No, no, I wasn't, I was a retail assistant at the time. SHE014

Repetitive activities were cited as the most frequent cause with many participants describing gardening or do-it-yourself (DIY) tasks as the precipitating factor. It may be argued that many of these are seasonal and would also constitute a sudden change in activity:

I was just trimming the hedges back in the garden, the physio said that's a classic—it's very common, repetitive vibration if you like and it can damage the tendon. SHE016

It came on while I was doing some sawing of an old crate for firewood. And I got carried away and did too much of it and then it hurt a little bit at the time and that's what started it. I never had a problem before. DER003

Others considered additional musculoskeletal conditions to have impacted on their ability to perform tasks normally, putting undue strain on their elbow and initiating their LET symptoms:

I think it's probably repetitive stress to be fair. I do a lot of DIY. I do most things myself so I think

it's just overuse. And I did also have carpal tunnel so whether the fact that, I've been operated on it now, but the fact that this was maybe another trigger or another factor that meant that I was maybe using my elbow not as I should be to try and compensate for my carpal tunnel, I'm sure. SHE018

Well, the reason why I have had to use the one with the Tennis Elbow, I think it started up because obviously I found that I had got arthritis in my shoulder, so they said try not to carry as heavy things with that arm. So obviously I have gone using the opposite arm, which is what the Tennis Elbow is in, and I think that's what's triggered it, because I have stopped using my right arm and I have started using my left arm. SHE001

## Theme 2: impact on everyday life

Participants reported that they continued to work but highlighted the difficulty in doing so, with some experiencing constant pain.

Well, basically the pain was constantly there especially when I was working, so yes it was in everyday life basically. BHX004

From a career perspective it was having a constant impact, constant pain, difficult to perform the necessary tasks and then on a day-to-day perspective I was having difficulty with things like carrying my shopping home. DER006

For many, the pain was severe enough to disturb their sleep—a characteristic often attributed to inflammation, infection or more sinister pathology but, historically in the literature, not usually associated with LET.

Sleeping is the worst; I virtually have to sleep with my arm like across my chest because I just can't get comfy. And if I lay on it, well obviously the pain it wakes me up and obviously then it takes me forever to get back to sleep. So, it's definitely disturbing my sleep. SHE005

One of the other factors has been at night it's been very uncomfortable; I tend to lay on that side just from a breathing perspective I end up on that side. And it's becoming very painful just moving my arm to turn over. That's probably been the worst of this particular bout of Tennis Elbow that I've had but that comes and goes. Sometimes, last November, December it was really painful at night, this last couple of months a bit better and then the last few nights have not been so good. SHE013

The majority of participants had found difficulty or significant pain when performing even the most basic of daily tasks such as washing and dressing, household chores, driving the car or even eating and drinking:

It's been quite major to be fair. Obviously, it's got to a stage where I can't really carry my shopping. I can't throw the ball for my dogs. I can't lift any heavy

weights. At one point I could barely pick up a kettle. I can't pick up cups of coffee. Anything with a weight to it I'm really struggling with. At one point I could I was struggling to put my car into reverse because it's a push down and push across motion and I couldn't do it. Yes, it's been quite major to be honest. SHE004

I couldn't really carry my shopping bags on my right arm anymore. Difficulty with basic, well at that point essential care things like putting my mask on [during the COVID-19 pandemic] when I was going outside or would be meeting with other people like simple gesture of raise your arm and twist, put the ear loop on really hurt by an astonishing amount for such a small gesture. Plus of course difficulty with little things like washing my hair in the shower. So, you know it was, it was persistent, it was there all parts of my life. It wasn't always painful so much as discomforting or just a bit limiting but you know it was quite a bother. DER006

Some had experienced episodes where their grip was weakened to such an extent that they were unable to hold objects:

I mean a few weeks ago I was pretty much in tears at the physiotherapy because—well, I'd been shopping and I'd picked up a two-litre bottle of pop and couldn't hold it and I dropped it and it smashed all over the floor. I had to find people to come and clean that up for me. Then I went round the next aisle and picked up a big bottle of bubble bath and did exactly the same thing. It was not a good day. I think the people at the supermarket was just glad to see the back of me that day, because I just kept dropping stuff and smashing it on the floor. SHE004

I mean I couldn't even paint a wall, cutting in was too painful, I ended up dropping the paintbrush. SHE018

Many gave examples of how the condition had impacted on the hobbies that they enjoyed with DIY tasks, gardening and exercise/gym featuring regularly.

Well, I was a keen gym-goer, loved keeping fit and that and when the pain started coming it started to restrict my movement in my arm, the speed I could do it. So, even though it didn't affect my actual workout, afterwards it would just be very painful, stop me from doing things. To the point where sometimes I had to stop using my right arm because of the pain and eventually it stopped me from going to the gym altogether. Which I didn't like, I hated not going to the gym but it stopped me from going to the gym eventually because I thought it needs to get better so, I needed to rest it. So, that's when I went to physiotherapy. BHX003

So, from the outset the main problem was doing DIY projects, any lifting, doing door handles. It's quite limiting … So, it's quite restricting, a lot of non-work activities that I normally do around the house and DIY in particular. DER003

### Theme 3: self-help and understanding of the condition

While some participants did search for self-help advice online, using Google or YouTube, there was hesitancy from many. Some were unsure of the diagnosis and therefore what to search for, and some did not want to follow potentially incorrect advice without seeking the opinion of a healthcare professional, for fear of making their symptoms worse.

I did Google once I was diagnosed as having Tennis Elbow. You're kind of curious to know what exactly it is, but I didn't do anything on the back of it. As in I didn't try to do anything myself because I didn't want to do anything wrong. DER011

As mentioned in the introduction, the diagnostic label of Tennis Elbow is widely used, despite calls for it to be referred to as LET. Several participants reflected that the name of the condition was misleading. They had assumed that only athletes or people with sporting hobbies could get Tennis Elbow and therefore it could not apply to them.

… but I didn't think it would be Tennis Elbow, so I didn't actually look up Tennis Elbow. But once it was pointed out, that was fairly obvious. You know, kind of a bit embarrassing. I didn't think that might have been a possibility but I mean the name is misleading, isn't it? … One always thinks sports. DER006

I knew the name but I didn't really think it was something—I assumed it was a sports injury from, you know, active physical sportsmen and I don't really do that. So, it was when they said it sounds like it was Tennis Elbow when I went to the doctors' I did some more research on it and then that sounded about right. As I say, it wasn't something, I knew the word, it wasn't something I expecting to be suffering from. SHE016

### Theme 4: healthcare experience

Participants had experienced a wide range of treatments via the public healthcare system and privately, prior to entering the OPTimisE pilot and feasibility trial. They had typically consulted their general practitioner initially and been offered corticosteroid injections, prescription painkillers or non-steroidal anti-inflammatory drugs (NSAIDs), often with just a short-term temporary effect.

I had a couple of injections; you know the injections they do for the steroids and stuff. That helped me. It went away for a couple of years. I didn't have any pain but after it came back again … Yes, basically I went back to the GP to the doctors and they suggested to have again a couple of injections. I had the first and then after three months another one. They didn't seem to have helped at all the last injections. BHX004

One participant experienced a rare but serious side-effect to oral NSAIDs, having not trialled topical versions. Acute eosinophilic pneumonia can be fatal but, in this case, fortunately the patient recovered following hospital admission and treatment.[12 13]

> In fact, I mean for me it was, the first time I went to the NHS was back in April and actually, he prescribed naproxen for me. Which the symptoms of the Tennis Elbow disappeared immediately within a couple of days. But I got an allergic reaction to it and developed eosinophilic pneumonia so, I was really in a really bad way for a few weeks. Had to have a six week course of steroids to clear that pneumonia so, that was pretty scary. SHE013

Another had received several different treatments for their other elbow previously and underwent surgery when their symptoms had failed to respond.

> Yes, I had physio. I went to see a sports massage therapist who did like laser on it, acupuncture. But it just got so bad that the doctor then referred me to have it operated on. SHE005

Some had been referred to physiotherapy and some had sought this privately with treatments including ultrasound therapy, massage, laser, acupuncture, shockwave therapy and exercise prescription. Many had been advised to purchase an elbow orthosis without any clear recommendation of which type to buy, often finding them ineffectual.

> I had to buy my own. So, she gave me a link to an App that had some information about Tennis Elbow … It had videos on it and some written information and amongst all of that was some pictures of Tennis Elbow supports. So, the first one that was suggested on there was the standard tubular bandage which I tried and that was immediately a disaster. I mean I couldn't find one that was the right size for me … So, after that I tried the second one which was the strap round the forearm … it was a plain white band that was meant to go just below your elbow. It had like a padded bit on it on the inside that was meant to go over the muscle that's attached to that tendon to sort of act as a pressure point to reduce the amount of action that muscle has got I guess. It didn't really have very good instructions or anything with it … but I had it checked by my physio and it was the right size and I was wearing it properly but it was causing me pain every time I was wearing it. DER006

Some were given the advice to completely avoid all painful activities.

> When I last went to the GP I said to her, I need to do some decorating at home. My house needs decorating I'm desperate. She says under no circumstances do any decorating, because that'll just aggravate it. Golf, you can't go and play golf. Don't go to the gym.

> Don't do any exercises at the gym, weight bearing or anything like that, because that will aggravate it and it'll put you back to square one. SHE004

> Yes, I saw the Community Physio. She diagnosed it and prescribed total rest. Do not use that arm she says, forget you've even got it. DER006

This may have resulted in some developing fear-avoidance behaviour.

> My biggest issue: I was afraid of doing too much and being in pain again. SHE014

## DISCUSSION

To our knowledge, this is the first qualitative study exploring the lived experience of people with LET. We identified four themes related to the experience of patients with LET: (1) known cause of onset, (2) impact on everyday life, (3) self-help and understanding of the condition and (4) the healthcare experience.

In line with previous evidence, many participants reported knowing what they felt had caused the onset of their symptoms, such as a sudden change of activity or repetitive use.[6 14 15] However, this needs to be treated with some caution due to known issues around patients misattributing the causes of their pain, as well as some patients relating the cause of decreased function to other comorbid health conditions in the affected arm, such as shoulder pain or carpal tunnel syndrome, which are commonly associated with LET.[16–18]

The study findings suggest that people with LET frequently experience a substantial impact on basic functional tasks and hobbies although most were able to continue to work regardless of the pain. It has previously been estimated that up to 5% of working-age adults with LET take time off work.[5] It was also common (around a third of our sample) for participants to describe sleep disturbance as a key impact from the condition. This is a novel finding. Previous estimates from a general population study suggest that disturbed sleep is rarely associated with LET.[4] It is likely that these differences relate to the current study recruiting patients who were actively seeking healthcare, rather than those from the general population, and who reported high levels of baseline pain and disability. According to the subgrouping proposed by Coombes et al, 5/17 of our participants had moderate baseline severity and 8/17 had high baseline severity.[19]

Participants reported feeling reluctant to seek self-help advice until they had received a diagnosis for their symptoms from a healthcare professional, for fear of doing self-administered treatment that may 'aggravate' their symptoms. During the OPTimisE intervention design, condition-specific and general health advice and education, including basic pain science, delivered by a physiotherapist and supported by printed and online resources, was selected as one of the three key treatment elements.[20] The intervention did include a greater focus on lifestyle

change, with resources made available to support physical re-activation and the adoption of healthy sleep and dietary habits, and social engagement where relevant. Interestingly, the lay description of 'Tennis Elbow' was unrelatable to many participants who did not lead active lifestyles. As the literature suggests that only about 5% of people with the condition actually play tennis,[21] it is perhaps time for this terminology to be removed from the clinical vocabulary to avoid confusion.

Many participants perceived that they had received healthcare treatments that were ineffectual and, in one instance, harmful. The National Institute for Health and Care Excellence's clinical knowledge summary advises initial management of activity modification, consideration of an orthosis and paracetamol or topical NSAID gel. If symptoms persist beyond 6 weeks, physiotherapy is recommended and oral NSAIDs can be considered if topical NSAIDs have failed to provide relief. It advises against routinely offering corticosteroid injections due to limited long-term effectiveness and high rate of symptom recurrence.[22] Some of our participants were frustrated by the lack of clarity on the type of orthosis to use. Others had received second-line interventions such as corticosteroid injections or oral NSAIDs as first-line interventions. Oral NSAIDs carry risk of gastrointestinal bleeding, myocardial infarction and stroke, as well as rarer side-effects.[23] In a review of hospital admissions for adverse drug reactions, 29.6% were found to have been caused by NSAID use, with some proving fatal.[24] One of our participants was hospitalised as a result of taking oral NSAIDs and their experience questions whether the practice of prescribing oral NSAIDs should continue when there is a lack of evidence to suggest that they are effective for LET.[25] There appears to be a risk that clinicians are offering patients false-hope by providing treatments that are not clinically effective. Alternatively, more focused guidance and reassurance that the condition is safe to live with and will resolve over time might be plausible, is worthy of further consideration and might help patients to avoid seeking unnecessary over-treatment. When the OPTimisE intervention was designed, a group of physiotherapists with special interest in LET, physiotherapy service managers and patients who had experienced LET formed a consensus on what an optimised physiotherapy protocol should include, based on best evidence and practicality.[20] This includes detailed education and advice, the provision of an orthosis and a progressive exercise regime. Certain treatments, such as ultrasound therapy, massage, acupuncture and shockwave therapy, were excluded. Interestingly, some participants in this qualitative study had received such treatments and found them to be ineffective. The OPTimisE intervention education and advice component includes activity modification, pacing, promotion of self-efficacy and basic pain science. A concern highlighted by the interviews therefore was the advice given to some to completely avoid all painful activities. Thoughts and beliefs that movement may cause structural tissue injury are known to result in increased levels of pain perception and contribute to the development of persistent pain.[26] It may also be a factor in why some patients fail to improve with treatment or fail to recover their full levels of function.

Strengths of this study are that this is the first qualitative study exploring the impact of LET on patients' everyday life and included individuals from a range of socioeconomic and ethnic backgrounds. We acknowledge though that qualitative studies draw on the opinions of those interviewed and may not be truly representative of the wider population; however, our aim was in-depth exploration of these views. The opinions presented are from people accessing healthcare for LET and so may not reflect the views of those who do not access healthcare for LET, due to lower symptom severity, stoicism or inaccessibility. This may exclude some underserved groups.

## CONCLUSION

Patients seeking healthcare for LET typically described: that they perceived the problem to be a form of damage from a specific activity; that the condition had a substantial impact on their ability to perform daily tasks, sleep, work and pursue hobbies; feeling hesitant to rely on online self-management information without first getting formal healthcare advice; that the common label of 'Tennis Elbow' was largely unrelatable for those who do not play sports; and that many of the treatment options they had received from healthcare providers were ineffective and offered false hope. Given uncertainty regarding the clinical and cost-effectiveness of current treatments for LET, this study provides stimulus for clinicians to consider the advice and treatment provided, and whether the messages conveyed reflect the favourable natural history and positive long-term outcome for many.

**Acknowledgements** The authors wish to thank all of the participants who gave their time for the interview. MB would like to thank the British Elbow & Shoulder Society for the award of a Research Pump-Priming Grant used to support preliminary study design and application for fellowship funding.

**Contributors** MB conducted the interviews and the transcript coding, with mentorship from BS. MB produced the initial codebook before MB, BS, KC and JCH collectively finalised the codebook and interpreted the themes. MB, BS, KC, CL and JCH contributed to this final written report. MB was responsible for the overall content as the guarantor.

**Funding** Marcus Bateman is funded by a National Institute for Health Research (NIHR) Chartered Society of Physiotherapy Charitable Trust Doctoral Fellowship (reference NIHR300704).

**Disclaimer** This paper presents independent research funded by the National Institute for Health Research (NIHR) and Chartered Society of Physiotherapy Charitable Trust. The views expressed are those of the author(s) and not necessarily those of Chartered Society of Physiotherapy Charitable Trust, the NHS, the NIHR or the Department of Health and Social Care.

**Competing interests** None declared.

**Patient and public involvement** Patients and/or the public were involved in the design, or conduct, or reporting, or dissemination plans of this research. Refer to the Methods section for further details.

**Patient consent for publication** Not applicable.

**Ethics approval** This study involves human participants and the approvals were received from the Yorkshire and The Humber—Sheffield Research Ethics Committee (reference 21-YH-0121) and the UK Health Research Authority (reference 297637) on 22 June 2021. Participants gave informed consent to participate in the study before taking part.

**Provenance and peer review** Not commissioned; externally peer reviewed.

**Data availability statement** Data are available upon reasonable request. A copy of the codebook is available on written request.

**ORCID iDs**
Marcus Bateman http://orcid.org/0000-0002-3203-506X
Jonathan C Hill http://orcid.org/0000-0001-6246-1409
Benjamin Saunders http://orcid.org/0000-0002-0856-1596

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
