## [Reviewer comments · BMJ Open]

ARTICLE DETAILS

TITLE (PROVISIONAL)	The Lived Experience of People with Lateral Elbow Tendinopathy – a Qualitative Study from the OPTimisE Pilot & Feasibility Trial.
AUTHORS	Bateman, Marcus; Hill, Jonathan; Cooper, Karin; Littlewood, Chris; Saunders, Benjamin

VERSION 1 – REVIEW

REVIEWER	Ring, David The University of Texas at Austin
REVIEW RETURNED	06-Feb-2023

GENERAL COMMENTS	1. The disease is an enthesopathy, not a tendinopathy. https://pubmed.ncbi.nlm.nih.gov/36563131/. I notice that the panel in reference 2 seems unaware of this. 2. Abstracts, objectives: “impact on quality of life” sounds quantitative. I would omit this and focus on the qualitative. 3. Abstract, setting: A qualitative study would be entirely separate from a randomized trial. What are you saying? 4. You have documented the cognitive errors typical of human illness behavior: a) New symptoms are often misinterpreted as new pathophysiology or an injury(1–4).b) Discomfort and incapability are strongly associated with unhelpful thoughts and feelings of distress(5–9).c) We often let people down by reinforcing unhelpful thinking(10,11).d) There is notable unwarranted variation in treatment(12). These issues are common to other MSK diseases and need to be handled with care(13). You seem to take what patients say as fact, which is discordant with best evidence. The lived experience of symptoms is characterized by misinterpretation (unhelpful thoughts) and disproportionate worry or despair(6,7). 5. Introduction, prevalence: The lifetime prevalence of enthesopathy of the ECRB origin is about 1 in 5(14). 6. You’ve catalogued human illness behavior. Now you need to interpret it correctly. People that present for symptoms related to enthesopathy of the ECRB origin experience greater discomfort and incapability in proportion to their thoughts and feelings about their symptoms. In other words, this qualitative work is in line with a large body of quantitative evidence. The way you present the information seems to suggest that people’s interpretation of their symptoms is accurate. That is a big step in the wrong direction.
--

	References  1. Lemmers M, Versluijs Y, Kortlever JTP, Gonzalez AI, Ring D. Misperception of Disease Onset in People with Gradual-Onset Disease of the Upper Extremity. J Bone Joint Surg Am. 2020 Dec 16;102(24):2174–80. 2. van Hoorn BT, Wilkens SC, Ring D. Gradual Onset Diseases: Misperception of Disease Onset. J Hand Surg Am. 2017 Dec;42(12):971-977.e1. 3. Liu TC, Leung N, Edwards L, Ring D, Bernacki E, Tonn MD. Patients Older Than 40 Years With Unilateral Occupational Claims for New Shoulder and Knee Symptoms Have Bilateral MRI Changes. Clin Orthop Relat Res. 2017 Oct;475(10):2360–5. 4. Furlough K, Miner H, Crijns TJ, Jayakumar P, Ring D, Koenig K. What factors are associated with perceived disease onset in patients with hip and knee osteoarthritis? J Orthop. 2021 Aug;26:88–93. 5. Lindenhovius A, Henket M, Gilligan BP, Lozano-Calderon S, Jupiter JB, Ring D. Injection of dexamethasone versus placebo for lateral elbow pain: a prospective, double-blind, randomized clinical trial. J Hand Surg Am. 2008 Aug;33(6):909–19. 6. Miner H, Rijk L, Thomas J, Ring D, Reichel LM, Fatehi A. Mental-Health Phenotypes and Patient-Reported Outcomes in Upper-Extremity Illness. J Bone Joint Surg Am. 2021 Aug 4;103(15):1411–6. 7. Teunis T, Al Salman A, Koenig K, Ring D, Fatehi A. Unhelpful Thoughts and Distress Regarding Symptoms Limit Accommodation of Musculoskeletal Pain. Clin Orthop Relat Res. 2022 Feb 1;480(2):276–83. 8. Das De S, Vranceanu AM, Ring DC. Contribution of kinesophobia and catastrophic thinking to upper-extremity-specific disability. J Bone Joint Surg Am. 2013 Jan 2;95(1):76–81. 9. Ring D, Kadzielski J, Fabian L, Zurakowski D, Malhotra LR, Jupiter JB. Self-reported upper extremity health status correlates with depression. J Bone Joint Surg Am. 2006 Sep;88(9):1983–8. 10. Goyal R, Mercado AE, Ring D, Crijns TJ. Most YouTube Videos About Carpal Tunnel Syndrome Have the Potential to Reinforce Misconceptions. Clin Orthop Relat Res. 2021 Oct 1;479(10):2296–302. 11. Drake ML, Ring DC. Enthesopathy of the Extensor Carpi Radialis Brevis Origin: Effective Communication Strategies. J Am Acad Orthop Surg. 2016 Jun;24(6):365–9. 12. Kachooei AR, Talaei-Khoei M, Faghfoury A, Ring D. Factors associated with operative treatment of enthesopathy of the extensor carpi radialis brevis origin. J Shoulder Elbow Surg. 2016 Apr;25(4):666–70. 13. Ulack C, Suarez J, Brown L, Ring D, Wallace S, Teisberg E. What are People That Seek Care for Rotator Cuff Tendinopathy Experiencing in Their Daily Life? J Patient Exp. 2022;9:23743735211069812. 14. van Leeuwen WF, Janssen SJ, Ring D, Chen N. Incidental magnetic resonance imaging signal changes in the extensor carpi radialis brevis origin are more common with age. J Shoulder Elbow Surg. 2016 Jul;25(7):1175–81.
--	--

REVIEWER	Letafatkar, Amir Kharazmi University, Department of Biomechanics and Sport Injuries
-----------------	--

REVIEW RETURNED	14-Feb-2023
-------------

GENERAL COMMENTS	Thank you for the opportunity to review the manuscript. The objective of the study was to explore the lived experience of people with Lateral Elbow Tendinopathy (LET) and its impact on quality-of-life. Some points need to be clarified, they are:  1. I recommend that after the abstract, in addition to "Strengths and limitations of this study", add the "Key messages" section. 2. Please describe why you did not include people who did not have access to healthcare for LET in the study? 3. In the introduction: I suggest that the introduction be rewritten, the introduction is short, and it seems that the importance of the topic is not well explained. 4. Please also add the "Aims and Hypotheses" section to the manuscript. 5. In the method section: I suggest using subtitles (such as: design, context, sample and recruitment, data collection and...). 6. I suggest you revise the 'Patient and public involvement' section according to the following questions: How was the development of the research question and outcome measures informed by patients' priorities, experience, and preferences? How did you involve patients in the design of this study? Were patients involved in the recruitment to and of the study? How will the results be disseminated to study participants? 7. Why are limitations and strengths discussed twice in the manuscript (in the last part of the discussion and after the abstract)? I suggest you address the limitations and strengths of the study in a subtitle, separate from the discussion; Or mention the points in the beginning section (after the abstract) 8. Please add the "SRQR Reporting checklist for qualitative study" checklist. I hope my comments can support the authors in doing. I admire them for choosing this topic. I wish them the best of luck with it.
---

VERSION 1 – AUTHOR RESPONSE

Reviewer: 1

Prof. David Ring, The University of Texas at Austin, Dell Medical School -- University of Texas, Austin
Comments to the Author:

1. The disease is an enthesopathy, not a tendinopathy.
<https://gbr01.safelinks.protection.outlook.com/?url=https%3A%2F%2Fpubmed.ncbi.nlm.nih.gov%2F36563131%2F&data=05%7C01%7Cmarcus.bateman%40nhs.net%7C74b5b7d73bde46347c4308db33b95f58%7C37c354b285b047f5b22207b48d774ee3%7C0%7C0%7C638160643478745975%7CUnknown%7CTWFpbGZsb3d8eyJWljoiMC4wLjAwMDAiLCJQIjoiV2luMzIiLCJBTiI6Ikl1haWwiLCJXVCi6Mn0%3D%7C3000%7C%7C%7C&sdata=4ZW2iZRD6ALHRNIObtJO5VGpv%2FZX3v7laplIT9rPO6M%3D&reserved=0>. I notice that the panel in reference 2 seems unaware of this.

We acknowledge the references provided below and have adjusted the introduction to include reference to the enthesis. The purpose of this paper is to explore patients' experience of living with the condition and therefore use the description of Lateral Elbow Tendinopathy as a label, as agreed by international consensus (reference 2 in manuscript). This will also aid library searches in future as the Medical Subject Heading (MeSH) is indexed under Elbow Tendinopathy.

2. Abstracts, objectives: “impact on quality of life” sounds quantitative. I would omit this and focus on the qualitative.

We have amended this to ‘everyday life’ to differentiate it from standardised quality of life measures.

3. Abstract, setting: A qualitative study would be entirely separate from a randomized trial. What are you saying?

As stated, this is a mixed-methods study (i.e. a combined qualitative and quantitative study). The main pilot and feasibility trial qualitative and quantitative results papers are forthcoming; this paper reports on one of the themes identified from qualitative interviews conducted as part of the trial.

4. You have documented the cognitive errors typical of human illness behavior:

a) New symptoms are often misinterpreted as new pathophysiology or an injury(1–4).

b) Discomfort and incapability are strongly associated with unhelpful thoughts and feelings of distress(5–9).

c) We often let people down by reinforcing unhelpful thinking(10,11).

d) There is notable unwarranted variation in treatment(12).

These issues are common to other MSK diseases and need to be handled with care(13). You seem to take what patients say as fact, which is discordant with best evidence. The lived experience of symptoms is characterized by misinterpretation (unhelpful thoughts) and disproportionate worry or despair(6,7).

The purpose of this study is to document the patient experience of living with the condition, as this has not been done before. Their perspectives can inform clinicians and are important for supporting the conversations clinicians have with patients, as well as potentially informing future self-management interventions. We are careful to use language, such as ‘perceived’, to describe the thoughts and feelings expressed by those interviewed. This study differs from previous studies using patient-reported outcome measure scores as it explores opinions outside of the constraints of a closed questionnaire. We have added the Drake & Ring (11) reference to the introduction to explain how this qualitative study may help clinicians to communicate with and reassure patients more effectively.

5. Introduction, prevalence: The lifetime prevalence of enthesopathy of the ECRB origin is about 1 in 5(14).

We refer to the point prevalence (i.e. how many people have the condition at a given point in time) in the adult general population of between 1.3 and 4.4%, rather than the lifetime prevalence. This is why the rate is lower than you describe in reference 14. The wording of this sentence in the introduction has been adjusted to clarify this.

6. You’ve catalogued human illness behavior. Now you need to interpret it correctly. People that present for symptoms related to enthesopathy of the ECRB origin experience greater discomfort and incapability in proportion to their thoughts and feelings about their symptoms. In other words, this qualitative work is in line with a large body of quantitative evidence. The way you present the information seems to suggest that people’s interpretation of their symptoms is accurate. That is a big step in the wrong direction.

This is where qualitative studies can be very valuable to clinicians, over-and-above what is known from quantitative studies. By highlighting the thoughts and concerns of patients, clinicians may have greater insight and be able to address those issues with simple advice. We also highlight in the discussion that pain-related fear can be a potential driver for central sensitisation resulting in a perception of higher levels of pain, as you describe.

References

1. Lemmers M, Versluijs Y, Kortlever JTP, Gonzalez AI, Ring D. Misperception of Disease Onset in People with Gradual-Onset Disease of the Upper Extremity. *J Bone Joint Surg Am.* 2020 Dec 16;102(24):2174–80.
2. van Hoorn BT, Wilkens SC, Ring D. Gradual Onset Diseases: Misperception of Disease Onset. *J Hand Surg Am.* 2017 Dec;42(12):971-977.e1.
3. Liu TC, Leung N, Edwards L, Ring D, Bernacki E, Tonn MD. Patients Older Than 40 Years With Unilateral Occupational Claims for New Shoulder and Knee Symptoms Have Bilateral MRI Changes. *Clin Orthop Relat Res.* 2017 Oct;475(10):2360–5.
4. Furlough K, Miner H, Crijns TJ, Jayakumar P, Ring D, Koenig K. What factors are associated with perceived disease onset in patients with hip and knee osteoarthritis? *J Orthop.* 2021 Aug;26:88–93.
5. Lindenhovius A, Henket M, Gilligan BP, Lozano-Calderon S, Jupiter JB, Ring D. Injection of dexamethasone versus placebo for lateral elbow pain: a prospective, double-blind, randomized clinical trial. *J Hand Surg Am.* 2008 Aug;33(6):909–19.
6. Miner H, Rijk L, Thomas J, Ring D, Reichel LM, Fatehi A. Mental-Health Phenotypes and Patient-Reported Outcomes in Upper-Extremity Illness. *J Bone Joint Surg Am.* 2021 Aug 4;103(15):1411–6.
7. Teunis T, Al Salman A, Koenig K, Ring D, Fatehi A. Unhelpful Thoughts and Distress Regarding Symptoms Limit Accommodation of Musculoskeletal Pain. *Clin Orthop Relat Res.* 2022 Feb 1;480(2):276–83.
8. Das De S, Vranceanu AM, Ring DC. Contribution of kinesophobia and catastrophic thinking to upper-extremity-specific disability. *J Bone Joint Surg Am.* 2013 Jan 2;95(1):76–81.
9. Ring D, Kadzielski J, Fabian L, Zurakowski D, Malhotra LR, Jupiter JB. Self-reported upper extremity health status correlates with depression. *J Bone Joint Surg Am.* 2006 Sep;88(9):1983–8.
10. Goyal R, Mercado AE, Ring D, Crijns TJ. Most YouTube Videos About Carpal Tunnel Syndrome Have the Potential to Reinforce Misconceptions. *Clin Orthop Relat Res.* 2021 Oct 1;479(10):2296–302.
11. Drake ML, Ring DC. Enthesopathy of the Extensor Carpi Radialis Brevis Origin: Effective Communication Strategies. *J Am Acad Orthop Surg.* 2016 Jun;24(6):365–9.
12. Kachooei AR, Talaei-Khoei M, Faghfour A, Ring D. Factors associated with operative treatment of enthesopathy of the extensor carpi radialis brevis origin. *J Shoulder Elbow Surg.* 2016 Apr;25(4):666–70.
13. Ulack C, Suarez J, Brown L, Ring D, Wallace S, Teisberg E. What are People That Seek Care for Rotator Cuff Tendinopathy Experiencing in Their Daily Life? *J Patient Exp.* 2022;9:23743735211069812.
14. van Leeuwen WF, Janssen SJ, Ring D, Chen N. Incidental magnetic resonance imaging signal changes in the extensor carpi radialis brevis origin are more common with age. *J Shoulder Elbow Surg.* 2016 Jul;25(7):1175–81.

Reviewer: 2

Dr. Amir Letafatkar, Kharazmi University Comments to the Author:

Thank you for the opportunity to review the manuscript. The objective of the study was to explore the lived experience of people with Lateral Elbow Tendinopathy (LET) and its impact on quality-of-life. Some points need to be clarified, they are:

1. I recommend that after the abstract, in addition to "Strengths and limitations of this study", add the "Key messages" section.

Thank you for the suggestion but the manuscript is currently formatted in accordance with the journal's guidance to only include a 'Strengths and limitations' section.

2. Please describe why you did not include people who did not have access to healthcare for LET in the study?

We have revised the 'Strengths and limitations' section to explain this.

3. In the introduction: I suggest that the introduction be rewritten, the introduction is short, and it seems that the importance of the topic is not well explained.

We have added further detail to the introduction.

4. Please also add the "Aims and Hypotheses" section to the manuscript.

The aims have been expanded in the introduction. Hypotheses are not relevant for this type of study.

5. In the method section: I suggest using subtitles (such as: design, context, sample and recruitment, data collection and...).

Subtitles have been added to the methods section, in line with those suggested

6. I suggest you revise the 'Patient and public involvement' section according to the following questions:

How was the development of the research question and outcome measures informed by patients' priorities, experience, and preferences? How did you involve patients in the design of this study? Were patients involved in the recruitment to and of the study? How will the results be disseminated to study participants?

More detail has been added to the PPIE section, as suggested.

7. Why are limitations and strengths discussed twice in the manuscript (in the last part of the discussion and after the abstract)?

It is a requirement of the journal to summarise the strengths and weaknesses after the abstract.

I suggest you address the limitations and strengths of the study in a subtitle, separate from the discussion; Or mention the points in the beginning section (after the abstract) 8. Please add the "SRQR Reporting checklist for qualitative study" checklist.

We have used the COREQ checklist, which is a similar checklist for reporting qualitative studies. It was provided with the original submission but may have not have been assimilated into the pdf review document.

VERSION 2 – REVIEW

REVIEWER	Ring, David The University of Texas at Austin
REVIEW RETURNED	02-May-2023

GENERAL COMMENTS	1. Discussion, end of second paragraph. Reference 16 identified associated factors. Those factors were essentially other diagnoses and treatments suggesting that at diagnosis of LET is associated with seeking care for other conditions. Data about claims (care seeking) can tell us nothing about factors associated with pathophysiology. They only tell us about feeling ill. People who feel ill enough to seek care with one condition are likely to seek care for another. But they are unlikely to be at greater risk for pathophysiology. 2. Discussion, paragraph 2. It is mandatory that you address that misattribution of cause is both common and associated with greater levels of discomfort and incapability. It is not reasonable to bypass this important evidence. 3. Discussion, paragraph 3. What you need to point out is that care-seeking and levels of discomfort and incapability have
---

	notable association with mindsets. Your language suggests that enthesopathy is always associated with notable illness. That is not true. People can accommodate it well. It's possible that most people with this condition never seek care. The people in your study were experiencing unhelpful thoughts, distress, lack of direction and also high enough levels of discomfort and incapability to seek care. But they are not likely representative of the average person with this pathophysiology. This is a key point that cannot be bypassed. 4. Discussion, paragraph 4. The idea of "aggravating" is a classic unhelpful thought. It would be so helpful to point out that your qualitative work uncovered the same things that the quantitative work has noticed: that the key to comfort and capability with LET is a healthy mindset. Don't miss this opportunity. 5. Discussion, paragraph 5. I think these themes are about false hope and lack of direction. Clinicians just need to be honest: "It's safe to live with and accommodate this condition in an active lifestyle, it will resolve without consequence and not return, and we've been searching for a way to get rid of it sooner, but we haven't found anything yet." This type of straight talk with compassion is what patients desire. Your evidence supports that. 6. Discussion, paragraph 6: I would stay away from the theory of central sensitization. You studied thoughts and feelings and there is a lot of evidence about how thoughts and feelings are associated with discomfort, incapability, and care-seeking behavior. Stick with that. 7. Here is my suggested rewrite of the Conclusion: "Patients with LET typically described notable discomfort and incapability. The themes identified were consistent with those identified in quantitative experiments: 1) A sense that the problem was a form of damage from specific activities, 2) Concern that additional activities could make the problem worse, and 3) Confusion regarding common labels and treatments that seem inappropriate and ineffective (false hope). The combined qualitative and quantitative data suggest that common words, concepts, and test and treatment strategies for LET may contribute to greater levels of discomfort, incapability, and frustration among people living with this condition. Clinicians can craft and practice effective communication strategies that help convey hope while sticking to the best evidence regarding enthesopathy."
--	---

VERSION 2 – AUTHOR RESPONSE

Reviewer: 1

Prof. David Ring, The University of Texas at Austin, Dell Medical School -- University of Texas, Austin
Comments to the Author:

1. Discussion, end of second paragraph. Reference 16 identified associated factors. Those factors were essentially other diagnoses and treatments suggesting that at diagnosis of LET is associated with seeking care for other conditions. Data about claims (care seeking) can tell us nothing about factors associated with pathophysiology. They only tell us about feeling ill. People who feel ill enough to seek care with one condition are likely to seek care for another. But they are unlikely to be at greater risk for pathophysiology.

The wording of this paragraph has been amended to remove the suggestion of risk associated with carpal tunnel syndrome or shoulder pain and also to address point 2, below.

2. Discussion, paragraph 2. It is mandatory that you address that misattribution of cause is both common and associated with greater levels of discomfort and incapability. It is not reasonable to bypass this important evidence.

The following statement has been included:

'However, this needs to be treated with some caution due to known issues around patients misattributing the causes of their pain, as well as some patients relating the cause of decreased function to other comorbid health conditions in the affected arm, such as shoulder pain or carpal tunnel syndrome, which are commonly associated with LET.'

3. Discussion, paragraph 3. What you need to point out is that care-seeking and levels of discomfort and incapability have notable association with mindsets. Your language suggests that enthesopathy is always associated with notable illness. That is not true. People can accommodate it well. It's possible that most people with this condition never seek care. The people in your study were experiencing unhelpful thoughts, distress, lack of direction and also high enough levels of discomfort and incapability to seek care. But they are not likely representative of the average person with this pathophysiology. This is a key point that cannot be bypassed.

Detail has been added to contextualise this point, in relation to the medium and high baseline severity reported by 13/17 of our population.

4. Discussion, paragraph 4. The idea of "aggravating" is a classic unhelpful thought. It would be so helpful to point out that your qualitative work uncovered the same things that the quantitative work has noticed: that the key to comfort and capability with LET is a healthy mindset. Don't miss this opportunity.

Thank you for the suggestion. Although, it is plausible to suggest that the key to comfort and capability with LET is a healthy mindset, we do not think this can be suggested with confidence. Although not reported here, the main findings from the OPTimisE pilot & feasibility trial were a favourable natural history in both intervention and usual care groups, irrespective of baseline levels of catastrophisation or self-efficacy. Hence, we have opted not to amend the manuscript in response to this point.

5. Discussion, paragraph 5. I think these themes are about false hope and lack of direction. Clinicians just need to be honest: "It's safe to live with and accommodate this condition in an active lifestyle, it will resolve without consequence and not return, and we've been searching for a way to get rid of it sooner, but we haven't found anything yet." This type of straight talk with compassion is what patients desire. Your evidence supports that.

You raise an important point, which is a belief we share, and likely to be the future direction of this project, but is not the focus of this paper. We have though added the following statement: 'There appears to be a risk that clinicians are offering patients false-hope by providing treatments that are not clinically effective. Alternatively, more focused guidance and reassurance that the condition is safe to live with and will resolve over time might be plausible, is worthy of further consideration and might help patients to avoid seeking unnecessary over-treatment.'

6. Discussion, paragraph 6: I would stay away from the theory of central sensitization. You studied thoughts and feelings and there is a lot of evidence about how thoughts and feelings are associated with discomfort, incapability, and care-seeking behavior. Stick with that.

Thank you we agree with this advice. This sentence has been reworded:

'Thoughts and beliefs that movement may cause structural tissue injury are known to result in increased levels of pain perception and contribute to the development of persistent pain.'

7. Here is my suggested rewrite of the Conclusion: "Patients with LET typically described notable discomfort and incapability. The themes identified were consistent with those identified in quantitative experiments: 1) A sense that the problem was a form of damage from specific activities, 2) Concern

that additional activities could make the problem worse, and 3) Confusion regarding common labels and treatments that seem inappropriate and ineffective (false hope). The combined qualitative and quantitative data suggest that common words, concepts, and test and treatment strategies for LET may contribute to greater levels of discomfort, incapability, and frustration among people living with this condition. Clinicians can craft and practice effective communication strategies that help convey hope while sticking to the best evidence regarding enthesopathy.”

Thank you. We have integrated some of these suggestions into the conclusion but feel that it is necessary to maintain the themes that we identified from the data, during the qualitative analysis:

‘Patients seeking healthcare for LET typically described: that they perceived the problem to be a form of damage from a specific activity; that the condition had a substantial impact on their ability to perform daily tasks, sleep, work and pursue hobbies; feeling hesitant to rely on online self-management information without first getting formal healthcare advice; that the common label of ‘Tennis Elbow’ was largely unrelatable for those who don’t play sports; and that many of the treatment options they had received from healthcare providers were ineffective and offered false hope. Given uncertainty regarding the clinical and cost-effectiveness of current treatments for LET, this study provides stimulus for clinicians to consider the advice and treatment provided, and whether the messages conveyed reflect the favourable natural history and positive long-term outcome for many.’

The abstract conclusion has also been updated.

VERSION 3 – REVIEW

REVIEWER	Ring, David The University of Texas at Austin
REVIEW RETURNED	08-Jun-2023

GENERAL COMMENTS	1. Abstract, Results, line one: Change to “Perceived cause of onset” 2. In theme 1, I believe you are missing an opportunity to reinforce the knowledge that people often misperceive new pains as injuries(1–3) and that doing so makes them less comfortable and capable(4–6). They perceive that they have damaged their arm. This is incorrect and unhealthy. “it didn’t feel right from the very first time I did it but I just carried on stupidly.” BHX003 “...because I’d gone from an office job to doing loads of shovelling over an intense period, I think I might have bumped my elbow as well while doing the job. Like I finished at the end of it and like my arm was screaming...” SHE011 “... it’s very common, repetitive vibration if you like and it can damage the tendon.” SHE016 “... I got carried away and did too much of it and then it hurt a little bit at the time and that’s what started it.” DER003 “I think it’s probably repetitive stress to be fair...that I was maybe using my elbow not as I should be to try and compensate for my carpal tunnel, I’m sure.” SHE018 3. Discussion, paragraph 2. There is evidence that people who perceive “overcompensation” or relate pains to new arm use are experiencing more unhelpful thinking(7,8). It seems appropriate to note that people who seek care for LET are experiencing important misconceptions and unhelpful thoughts.
--

	4. Discussion, paragraph 3. Sleep disturbance is associated with stress, distress, and unhelpful thinking. There's pattern here and it would be a shame to overlook it. 5. Discussion, paragraph 4. The concept of "aggravation" is also related to unhelpful thinking. 6. The conclusion paragraph is brilliant. References  1. Lemmers M, Versluijs Y, Kortlever JTP, Gonzalez AI, Ring D. Misperception of Disease Onset in People with Gradual-Onset Disease of the Upper Extremity. J Bone Joint Surg Am. 2020 Dec 16;102(24):2174–80. 2. Liu TC, Leung N, Edwards L, Ring D, Bernacki E, Tonn MD. Patients Older Than 40 Years With Unilateral Occupational Claims for New Shoulder and Knee Symptoms Have Bilateral MRI Changes. Clin Orthop Relat Res. 2017 Oct;475(10):2360–5. 3. Furlough K, Miner H, Crijns TJ, Jayakumar P, Ring D, Koenig K. What factors are associated with perceived disease onset in patients with hip and knee osteoarthritis? J Orthop. 2021 Aug;26:88–93. 4. Das De S, Vranceanu AM, Ring DC. Contribution of kinesophobia and catastrophic thinking to upper-extremity-specific disability. J Bone Joint Surg Am. 2013 Jan 2;95(1):76–81. 5. Lindenhovius A, Henket M, Gilligan BP, Lozano-Calderon S, Jupiter JB, Ring D. Injection of dexamethasone versus placebo for lateral elbow pain: a prospective, double-blind, randomized clinical trial. J Hand Surg Am. 2008 Aug;33(6):909–19. 6. Teunis T, Al Salman A, Koenig K, Ring D, Fatehi A. Unhelpful Thoughts and Distress Regarding Symptoms Limit Accommodation of Musculoskeletal Pain. Clin Orthop Relat Res. 2022 Feb 1;480(2):276–83. 7. Romere C, Ramtin S, Nunziato C, Ring D, Laverty D, Hill A. Is Pain in the Uninjured Arm Associated With Unhelpful Thoughts and Distress Regarding Symptoms During Recovery From Upper-Extremity Injury? J Hand Surg Am. 2023 May 17;S0363-5023(23)00170-3. 8. Romere C, Ramtin S, Nunziato C, Ring D, Laverty D, Hill A. Is Pain in the Uninjured Leg Associated With Unhelpful Thoughts and Distress Regarding Symptoms During Recovery From Lower Extremity Injury? Clin Orthop Relat Res. 2023 May 29
--	---